# The Use of Sentinel Lymph Node Mapping for Canine Mast Cell Tumors

**DOI:** 10.3390/ani14071089

**Published:** 2024-04-03

**Authors:** Marta Romańska, Beata Degórska, Katarzyna A. Zabielska-Koczywąs

**Affiliations:** Department of Small Animal Diseases and Clinic, Institute of Veterinary Medicine, Warsaw University of Life Sciences, Nowoursynowska 159c, 02-776 Warsaw, Poland; marta_romanska@sggw.edu.pl (M.R.); beata_degorska@sggw.edu.pl (B.D.)

**Keywords:** blue dye, contrast-enhanced ultrasonography, dogs, indirect lymphography, mast cell tumors, sentinel lymph node mapping, near-infrared lymphography, lymphoscintigraphy

## Abstract

**Simple Summary:**

Canine mast cell tumors are cutaneous and subcutaneous tumors that spread through the lymphatic system and lead to high mortality rates. Lymph node mapping, crucial for pinpointing the sentinel lymph node—the initial recipient of lymph from a tumor—is pivotal in diagnosing cancers that spread through the lymphatic system. Lymph node mapping is critical for assessing disease progression, guiding treatment, and predicting outcomes. Furthermore, the early detection of lymph node metastases is essential for improving prognosis. Lymph node mapping is a routinely applied diagnostic technique in humans but a relatively new technique in veterinary oncology; only recently have the first studies been conducted on its application in mast cell tumors. The primary aim of this article is to summarize the current knowledge on various lymph node mapping methods in canine mast cell tumors, highlighting their advantages, disadvantages, and the importance of this approach both for veterinary practitioners and dog owners. The sentinel lymph node exploits unique drainage patterns in mast cell tumors, highlighting the role of lymph node mapping in the precise diagnosis and treatment of canine mast cell tumors; the successful diagnosis of mast cell tumors could lead to significant progress in veterinary oncology.

**Abstract:**

Cancer is the leading cause of death in companion animals. The evaluation of locoregional lymph nodes, known as lymph node mapping, is a critical process in assessing the stage of various solid tumors, such as mast cell tumors (MCTs), anal gland anal sac adenocarcinoma, melanoma, and mammary gland adenocarcinoma. MCTs are among the most prevalent skin malignancies in dogs. Staging is used to describe the extent of neoplastic disease, provide a framework for rational treatment planning, and evaluate treatment results. The aim of this review is to present the current knowledge on sentinel lymph node (SLN) mapping in canine MCTs, its influence on treatment decisions and prognosis, as well as the advantages and limitations of different SLN techniques currently available in veterinary oncology. A search methodology was adopted using the PubMed, Scopus, and Google Scholar databases. Critical analyses of up-to-date research have shown that lymphoscintigraphy can achieve a lymph node detection rate of between 91 and 100%. This method is becoming increasingly recognized as the gold standard in both human and veterinary medicine. In addition, initial studies on a limited number of animals have shown that computed tomographic lymphography (CTL) is highly effective in the SLN mapping of MCTs, with detection rates between 90 and 100%. The first study on contrast-enhanced ultrasound (CEUS) also revealed that this advanced technique has up to a 95% detection rate in canine MCTs. These methods provide non-ionizing alternatives with high detection capabilities. Furthermore, combining computed tomography and near-infrared fluorescence (NIR/NIR-LND) lymphography is promising as each technique identifies different SLNs. Indirect lymphography with Lipiodol or Iohexol is technically feasible and may be also used to effectively detect SLNs. The integration of these mapping techniques into routine MCT staging is essential for enhancing the precision of MCT staging and potentially improving therapeutic outcomes. However, further clinical trials involving a larger number of animals are necessary to refine these procedures and fully evaluate the clinical benefits of each technique.

## 1. Introduction

Cancer is the leading cause of death in companion animals, including domestic dogs [1]. Staging is used to describe the extent of neoplastic disease, provide a framework for rational treatment planning, and evaluate treatment results. Accurate staging requires an understanding of the biologic behavior of tumors [2]. The evaluation of locoregional lymph nodes, known as lymph node mapping, is a critical process in assessing the stage of various solid tumors, such as mast cell tumors (MCTs), anal gland anal sac adenocarcinoma, melanoma, and mammary gland adenocarcinoma [3,4].

MCTs represent a prevalent neoplasm in dogs, constituting approximately 20% of all canine integumentary tumors, making them the second most commonly diagnosed cancer in dogs, followed by mammary tumors [2,5,6,7]. The age of diagnosis is diverse, spanning from 7 months to 18 years, with an average age of onset of 8.2 years [6]. Predominantly affected breeds include Boxers, Labrador Retrievers, American Staffordshire Terriers, Golden Retrievers, French Bulldogs, Dachshunds, and Shar-Peis. Specifically, Boxers showcase a 96.8% predisposition to low-grade MCTs. Gender differences are also noteworthy; females, especially in the older age group (11–16 years), display an elevated risk for high-grade tumors. In contrast, younger dogs, particularly between 4 and 6 years old, are at a higher risk of developing low-grade tumors [8]. Within round cell tumors, MCTs are known to spread first to the lymph nodes; therefore, they are an excellent model for testing sentinel lymph node (SLN) mapping techniques in dogs [9]. The SLN is the first lymph node that drains the lymph of the primary tumor. The use of SLN mapping has become a gold standard in human medicine for breast cancer, endometrial cancer, and melanoma [3,10,11]. The SLN can be a predictor of metastatic disease as it can be the first site where metastasis occurs [9].

MCTs are typically categorized into two primary types, cutaneous and subcutaneous, depending on their location. Cutaneous MCTs often emerge as single nodules on the skin, primarily involving the dermis. They often extend to the epidermis, leading to significant ulceration, and may also infiltrate the subcutaneous layer [7]. Subcutaneous MCTs, strictly confined to the subcutaneous region and encased in adipose tissue, are categorized into three growth patterns by Thompson et al.: circumscribed, combined (infiltrative/circumscribed), and infiltrative [12,13]. The initial diagnosis of the majority of MCTs is achieved using fine-needle aspiration. However, only surgical biopsies and histopathological analysis enable the distinction between cutaneous and subcutaneous MCTs and provide accurate tumor classification, which is critical for the animal’s prognosis [7]. Subcutaneous MCTs are typically characterized by less aggressive behavior than cutaneous ones. One of the causes of the less aggressive behavior of subcutaneous MCTs compared to cutaneous MCTs is thought to be related to the endocrine activity surrounding adipose tissue, which produces adipokines that modulate disease progression and mast cell maturation, potentially leading to a milder course of disease [13]. However, there are some reports with contrasting results [2], and there is a lack of studies that make direct comparisons between the behavior of cutaneous and subcutaneous MCTs [13]. 

Histological characteristics—such as distinct patterns and the mitotic index (MI: the ratio between the number of cells undergoing mitosis and the total number of cells in a population), together with the assessment of proliferation markers, such as Ki-67 (a nuclear protein highly expressed in cycling cells)—are essential for evaluating the aggressiveness of all types of MCTs, including subcutaneous MCTs. This is because histological grading criteria are not applicable for categorizing subcutaneous MCTs [14,15].

The staging and grading of cutaneous MCTs are instrumental for therapeutic decision making. Traditional staging techniques, such as lymph node assessments, abdominal ultrasound, and chest radiography, offer insights into the variability of MCTs [16]. Hume et al. [17] revealed that lymph node assessments and the control of locoregional MCTs significantly influence outcomes. Dogs with an early-stage diagnosis of grade 3 MCTs treated with adequate local control (ALC) experience longer survival times compared to those not treated with ALC, with the entire population showing a median progression-free survival (PFS) of 133 days and overall survival (OS) of 257 days. Specifically, for dogs without lymph node metastasis, median PFS was extended to 349 days and OS to 503 days, highlighting the prognostic importance of nodal involvement. In contrast, dogs with lymph node metastasis had a median PFS of 77 days and OS of 176 days. Lymph node status serves as a crucial prognostic factor, emphasizing the importance of appropriate metastatic lymph node treatment for improved survival [17].

The evaluation of the lymph node status is critical for the proper staging of MCTs. It should be remembered that the regional lymph node (RLN) (the lymph node anatomically closest to the tumor) may differ from the SLN due to complexity in the lymphatic chain draining the tumor [17,18,19,20,21]. The identification and assessment of the lymph node that directly drains the lymph from the tumor, expected to be the first metastatic lymph node, is essential for accurate prognosis [9,19,22]. Histopathological grading, however, is paramount for understanding the biological behavior of MCTs, thus influencing treatment pathways, including targeted therapies [16]. Two primary grading systems are currently used: the Patnaik Grading System and the Kiupel System.

### 1.1. The Patnaik Grading System

The Patnaik Grading System categorizes MCTs into three grades based on criteria such as cellular morphology, MI, cellularity, extent of tissue involvement, and stromal reaction. Grade 1 MCTs exhibit well-differentiated mast cells with no mitotic figures and minimal stromal reaction. Grade 2 MCTs show moderate-to-highly cellular structures, with mast cells displaying moderate pleomorphism and 0–2 mitotic figures per high-power field. Grade 3 MCTs are characterized by highly pleomorphic mast cells, 3–6 mitotic figures per high-power field, and extensive tissue invasion. Despite being commonly applied, this system is often criticized due to the challenges in distinguishing between grades, especially grade 1 and 2, leading to variability in pathologists’ interpretations [8,23,24]. Kiupel et al. highlight the necessity for the introduction of more precise classification systems due to current ambiguity, whereas Berlato et al. address the difficulties in predicting the behavior of grade 2 MCTs, with the majority having a benign clinical course, while approximately 20% exhibit aggressive behavior [25,26].

### 1.2. The Kiupel Grading System

The Kiupel System, a more recent classification method used for MCTs, categorizes these neoplasms into low-grade [Kiupel low-grade: (K-LG)] and high-grade [Kiupel high-grade: (K-HG)] tumors based on specific cellular features. These features include the degree of cellular differentiation, mitotic count, extent of cellular pleomorphism, and presence of multinucleated giant cells. Due to its straightforward approach, the Kiupel System has garnered significant acclaim among pathologists for ensuring reduced interpretation variability [8,26].

### 1.3. The Clinical Staging System

The World Health Organization (WHO) has established a clinical staging system for canine cutaneous MCTs to improve diagnostic and treatment accuracy. Detailed in Table 1, this system categorizes MCTs based on tumor location and size, lymph node involvement, distant metastasis, and the animal’s overall health. According to the WHO, dogs with stage I tumors generally exhibit longer survival rates compared to those with more advanced stages. It has been suggested that dogs harboring multiple tumors, thus categorized into stage III or IV, tend to have a worse prognosis. However, this approach, particularly for MCT grade 3, has been debated in various studies [16,27,28].

The assessment of lymph node status has fundamentally changed over the past two decades. Initially, this evaluation was limited to the physical examination of RLNs, and then it developed into analyzing the cytology and histopathology of RLNs and more recently to evaluating the first draining lymph node: the SLN. The histologic evaluation of the SLN is an important staging tool for assessing disease progression and making treatment decisions [29].

In human medicine, the status of the lymph node is a key prognostic factor in the evaluation of various cancers such as breast cancer and melanoma [30,31,32]. Furthermore, the resection of metastatic nodes is shown to enhance treatment outcomes for endometrial cancer, carcinoma of the tongue, and gastric cancer [3]. Until the mid-20th century, it was believed that lymph nodes located closest to the tumor mass were the initial sites of metastasis; therefore, radical surgeries involving the wide resection of the tumor and surrounding tissue, as well as the resection of the entire regional lymphatic basin, were standard procedures [33]. The approach to the radical resection of RLNs shifted with the development of sentinel lymph node (SLN) mapping techniques, particularly in breast cancer, melanomas, and cervical cancers. The advancement of SLN mapping and more selective lymphadenectomies has altered the approach to treating several cancer types, including malignant melanoma, breast cancer, cervical cancer, vulvar cancer, and stomach cancer [3,10,29]. In veterinary medicine, the first approaches to utilize SLN mapping were performed for various neoplasia, including MCTs [3,34]. 

The aim of this review is to present the current knowledge on SLN mapping in canine MCTs, its influence on treatment decisions and prognosis, and the advantages and disadvantages of the different SLN techniques currently available in veterinary oncology.

## 2. Search Methodology

The first literature search was conducted on the 19th of October 2023 using the PubMed, Scopus, Google Scholar, and Web of Science databases to identify pertinent studies on canine mastocytoma and lymph node mapping, initially utilizing the follow key search queries: ‘(canine mast cell tumour) AND (sentinel lymph node)’; ‘(canine mast cell tumor) AND (lymph node mapping)’, or ‘(dog mast cell tumor) AND (lymph node mapping)’. Each key search query yielded approximately 15–16 relevant results. A key focus was to ensure that at least 50% of the citations were from the last 5 years. Throughout the writing process, databases were repeatedly accessed to acquire newly published studies, and their abstracts were manually reviewed for information on lymph node mapping techniques, ensuring the inclusion of the most recent and relevant literature (with the last search conducted on 30 January 2024). Two authors performed an independent review of the related abstracts (M.R. and K-Z.K.). No language restrictions were applied to the search, but only articles in English were evaluated.

## 3. SLN Mapping Techniques for Canine MCTs

In recent years, a variety of mapping techniques for MCTs have emerged, each offering unique advantages in terms of efficacy and invasiveness. These techniques include the following: lymphoscintigraphy, colorimetric SLN mapping (utilizing the peritumoral injection of blue dye or indocyanine green), radiographic lymphography (also known as indirect lymphography or radiographic indirect lymphangiography), computed tomography lymphangiography (CTL), near-infrared fluorescence/near-infrared fluorescent image-guided lymphography (NIR/NIR-LND) and contrast-enhanced ultrasound (CEUS), and the techniques employed in [3,5,11,29,35]. All of these techniques are minimally invasive compared to radical lymphadenectomy, with diagnostic rates ranging between 77,9% and 100% depending on the technique used (Table 2).

SLN mapping techniques, often referred to as indirect due to the marker’s uptake by lymphatic vessels, involve administering a specific marker/dye/contrast agent near the tumor site using the four-quadrant method. The method involves the peritumoral administration of a specific marker, dye, or contrast agent at four different points surrounding the tumor. Direct intratumoral injections should be avoided. After administration, the marker is taken up by lymphatic vessels that drain the tumor mass, eventually reaching single or multiple SLNs [11,29].

### 3.1. Lymphoscintigraphy

Lymphoscintigraphy is based on the use of radiolabeled colloids (mainly technetium-99 m), preoperative planar imaging (two-dimensional visualization technique), and the intraoperative use of a hand-held gamma probe [3]. The isotope-labeled colloids are administered either intradermally or subcutaneously at the tumor site and stay there for a prolonged period. The uptake duration of these radioactive colloids is influenced by the colloid molecule size and the injected volume [3]. Notably, the isotope is primarily sequestered by the SLN and seldom progresses to second-echelon lymph nodes, which is a distinctive advantage of this method. Intraoperatively, a gamma camera aids in detecting a “hot” SLN, implying its absorption of the radiolabeled tracer [29]. This technique’s efficiency is further affirmed by its detection rate, which oscillates between 83 and 95% (depending on the type of tumor), rendering it the current gold standard in lymph node mapping [9,36,37]. For various head and neck tumors, including MCTs, the detection rate reported by Chiti et al. was 83% [36]. In their study, the SLN was detected in all MCTs (100% detection rate in eight out of eight MCTs) regardless of whether they were cutaneous (n = 4), subcutaneous (n = 2), or mucous membrane (n = 2) MCTs. In two cases, radioactive SLNs were identified but did not stain with blue dye. Conversely, in the other five cases, SLNs stained with blue dye were also detected via a radiocolloid [36].

The study by Ferrari et al. (2020) demonstrated a 91% MCT detection rate (31 out of 34 MCTs were detected) using combined lymphoscintigraphy and methylene blue, underlining the efficacy of SLN mapping in a cohort of 30 dogs [9]. This study included twenty-four cutaneous MCTs, six subcutaneous MCTs, three scars from the incompletely resected tumors, and one undefined tumor. The SLNs were not identified in three scar tissues, which suggests that the method might be less effective for recurrent or incompletely resected tumors [9]. The histological examination of the extirpated SLNs revealed metastasis in 56% of cases. This method proved effective, particularly for detecting SLNs in dogs with MCTs without prior surgical interventions, indicating the technique’s limitations in the presence of scar tissue [9]. Worley (2014) obtained even better results in a study of 19 dogs with 20 MCTs (undefined type), where 18 had SLNs preoperatively identified using regional lymphoscintigraphy. This approach preoperatively ascertained SLNs in 94.7% of the cases and intraoperatively achieved a 100% identification rate using the gamma probe. This high success rate was attributed to the combined use of lymphoscintigraphy and intrasurgical methylene blue injection, highlighting the potential of integrating various techniques for enhanced SLN detection in the management of MCTs [18]. Importantly, in 42.1% of these cases (8 out of 19 dogs), the identified SLNs were different from the anticipated RLNs. Additionally, all 19 dogs exhibited ‘hot’ SLNs detected by the gamma probe, and three dogs presented unexpected metastatic SLNs. This study underscores the enhanced accuracy and potential of lymphoscintigraphy combined with methylene blue to identify SLNs, particularly in cases of MCTs [18]. Manfredi et al. (2021) [37] reported a 95% detection rate of SLNs across various tumor types in a study involving 59 tumors in dogs, which included 47 cases of MCTs (undifferentiated between cutaneous and subcutaneous), alongside other tumor types. Specifically, the detection rate for MCTs in this study was notably high (97.9%), with SLNs successfully identified in 46 out of 47 MCT cases. The only failure to detect SLNs in MCTs occurred in a case of a recurrent MCT. Despite the overall high detection rate, the radiotracer failed to identify SLNs in three cases, including the SLN with a recurrent MCT and two cases of thyroid carcinoma [37].

Furthermore, the research conducted by Ferrari et al. involving 53 dogs with 66 MCTs (50 cutaneous and 16 subcutaneous) demonstrated a possible correlation between the size of low-grade cutaneous and subcutaneous MCTs and SLN status, showing that cMCTs equal to or larger than 3 cm and scMCTs had a higher risk of early or overt metastases compared to smaller tumors, indicating the effectiveness of the SLN mapping procedure in those tumors [21]. This finding complements the results of Chalfon et al., who demonstrated in their multivariable analysis that tumor diameter was the only variable significantly associated with an increased risk of local recurrence. Dogs with tumors larger than 3 cm exhibited a higher risk of both local recurrence and nodal metastasis, regardless of histological margins [42]. The insights from Ferrari et al. emphasize that, while smaller tumors may have a reduced likelihood of SLN metastasis, the decision regarding SLN mapping and biopsy should not solely rely on tumor size. A comprehensive approach, considering tumor size, type, degree of malignancy, and clinical symptoms, is essential in the management of MCTs in dogs. Therefore, decisions regarding SLN mapping and biopsy should be made on a case-by-case basis, integrating all clinical aspects to ensure a tailored and effective treatment strategy.

In a retrospective study, Gariboldi et al. evaluated 33 dogs presenting with 34 surgical scars from previous tumor excisions, including 29 MCTs, two soft tissue sarcomas, one oral melanoma, and one mammary adenocarcinoma. SLN biopsies were conducted, on average, 50 days after primary tumor removal. The study achieved a 100% SLN detection rate in the MCT group, with a 50% rate of SLN metastasis (combining HN2, early metastasis, HN3, and overt metastasis [4]). Two mapping techniques were employed: lymphoscintigraphy and NIR imaging. Lymphoscintigraphy utilized methylene blue combined with a radiopharmaceutical used for intraoperative SLN identification. For NIR lymphography, a specialized camera system (IC-FlowTM or Visionsense VS3 IridiumTM) was used to intraoperatively identify the draining lymphatic ducts and nodes, which were then dissected and excised under fluorescent guidance. Lymphoscintigraphy and NIR lymphography were utilized for the mapping of seventeen MCTs and ten MCT scars, and two MCT scars were mapped using both techniques. A significant limitation of the study was the absence of long-term follow-up, which hindered the assessment of false-negative rates. False negatives, identified as nodal metastases found in secondary or tertiary lymph nodes during follow-up without previous evidence of SLN metastasis, critically impact SLN resection accuracy. Moreover, the study noted that complex skin reconstructive surgeries, particularly rotational flaps, could significantly alter lymphatic drainage, resulting in a higher false-negative rate. This was especially noted in animals who had undergone such surgeries as their initial treatment [4].

In summary, lymphoscintigraphy stands as a potent tool in MCT lymph node mapping, with its efficacy supported by high detection rates. However, the fact that SLN detection was unsuccessful in the case of MCT recurrence underlines the need for further investigation into the technique’s efficacy in recurrent MCTs [37], as well as assessing the influence of MCT types on outcomes. Furthermore, SLN detection’s reliance on radioactive isotopes leads to challenges such as scarce equipment, elevated gamma probe expenses, and the inherent risks of radioactive work environments [3]. Nevertheless, lymphoscintigraphy is currently a gold standard for SLN and MCT detection both in human and veterinary medicine [9,34].

### 3.2. Indirect Lymphography

The indirect lymphography technique involves the use of either water-soluble contrast agents (e.g., Iopamidol, Omnipaque) or a lipid-soluble contrast agent (e.g., Lipiodol- iodinated ethyl esters from fatty acids of poppyseed oil), along with regional radiographs, to visualize SLNs. For this procedure, the selected contrast agent is injected near the tumor site using the four-quadrant technique. When Lipiodol is used (1 mL), regional radiographs are performed 24 h after injection. In contrast, when a water-soluble agent such as Iopamidol or Omnipaque is utilized (1–4 mL), radiographs are taken 3.5 min post injection and are repeated every 3 min (ranging from 1 to 18 min) until the contrast uptake of the lymphatic vessels occurs and SLNs become visible [3,5,39,43]. Injections with volumes of less than 4 mL have been shown to yield inconsistent results. According to Haas and collaborators, cases injected with 4 mL of contrast agent were associated with higher-quality imaging, especially when performed immediately after the injection [39]. Based on Annoni et al. research the minimum dose of the administered contrast agent was 1 ml for each 1 cm^2^ of the tumor base [40]. 

In the study on fifty-three dogs with a total of fifty-nine MCTs (undefined for the most part, with just seventeen histopathologically diagnosed as cutaneous MCTs and four identified as subcutaneous MCTs), which employed the technique of indirect lymphography using an iodinated water-soluble contrast medium (Omnipaque), the diagnostic detectability rate of SLN was 77.9%, and 22% of the observations were deemed non-diagnostic [39]. Interestingly, two out of four subcutaneous MCTs had metastatic lymph nodes upon histopathology diagnosis. Although this is a small number of subcutaneous tumors, these findings underscore the importance of SLN mapping, despite the previously reported less aggressive biologic activity of subcutaneous MCTs [39]. For cases considered diagnostic or partially diagnostic (where lymphatics are identifiable but the SLN is not highlighted), contrast uptake occurred on average within 3.5 min (ranging from 1 to 18 min), and imaging was typically concluded around 24 min after the injection (with a range of 8–90 min) with 3–4 mL of radiopaque medium [39].

The results of the study performed by De Bonis et al., encompassing twenty-six dogs with twenty-nine MCTs (twenty-one cutaneous and eight subcutaneous) that underwent indirect lymphography with Lipiodol, demonstrated variability in lymphatic drainage patterns as seventeen dogs were associated with a single SLN, while nine presented multiple SLNs [5]. Of the thirty-seven identified SLNs, 32% were confirmed as metastatic, based on either histological or cytological examinations [5]. The study achieved a commendable 89.6% success rate in detecting SLNs in MCTs as radiographic indirect lymphangiography with Lipiodol detected at least one SLN in 26 out of 29 primary tumors. Three undetected SLNs were for two subcutaneous and one cutaneous MCT, resulting in 87.5% and 95.6% detection rates, respectively. The difference in detection rates may be caused by the small number of subcutaneous cases included in the study. The procedure involved administering a relatively low volume of Lipiodol (0.8–1.6 mL), determined by tumor size, followed by a radiographic evaluation after 24 h. An example radiograph of the results of indirect lymphography using Lipiodol is shown on Figure 1. However, the research presented by De Bonis et al. has its limitations, notably the relatively small sample size and occasional reliance on cytology, which may not be as definitive as histopathology. Additionally, the absence of intraoperative verification methods adds a layer of uncertainty to SLN identification. While promising, this technique warrants further exploration with more expansive research parameters to solidify its applicability [5].

The dual method approach combining the preoperative injection of iodized oil around the primary tumor for indirect lymphography, followed by surgical excision of the SLNs after the peritumoral injection of methylene blue, performed on 29 dogs with 30 palpable tumors, including 15 MCTs (nine cutaneous, three subcutaneous, three scars from incompletely resected cutaneous MCTs), resulted in a 96.6% success rate, with a substantial 84.6% agreement observed for indirect lymphography and methylene blue techniques [38]. The study highlighted the significant role of methylene blue in improving the intraoperative visualization of SLNs, thereby contributing valuable insights to the fields of veterinary oncology and SLN mapping [38]. However, as noted in a later study by Ferrari et al., the use of methylene blue may have a disadvantage, especially in small tumors, where the dye may obscure the visualization of the deep fascial plane during stratification, posing a challenge to the complete resection of MCTs [9].

Hlusko and collaborators compared indirect lymphography with lymphoscintigraphy on eight healthy dogs [43]. When performing SLN mapping, lymphoscintigraphy achieved a 100% detection rate (eight out of eight dogs), while indirect lymphography achieved an 87.5% detection rate (seven out of eight dogs). For lymphography, 4 mL of 350 mg/mL Iohexol (Omnipaque, GE Healthcare, Chicago, IL, USA) was used, injected into the subcutaneous tissues using the four-quadrant technique, with the dose volume evenly divided among the four quadrants. Radiograms were performed at intervals of 0, 1, 2, 5, and 10 min post injection or until the SLN took up the contrast. However, the interpretation of these results should be approached with caution due to the limited number of animals included in the study [43].

In conclusion, indirect lymphography using an iodinated aqueous contrast medium demonstrates potential as an SLN mapping technique for MCTs in dogs. The broader applicability of radiographic indirect lymphangiography is emphasized by its feasibility, accessibility, minimally invasive nature, and cost-effectiveness compared to other mapping techniques. Despite the potential drawback of a 24 h waiting period post-Lipiodol administration, which prolongs the procedure relative to other methods, significant adverse effects related to Lipiodol have not been reported in the literature. It is generally safe with only mild peritumoral, post-injection swelling noted in some cases. Frequently, the sedation of the animal is necessary while performing the procedure. The challenges of indirect lymphography include the timing and volume of the contrast agent for optimal visualization. This technique requires further refinement in establishing optimal imaging intervals and determining the most suitable point to conclude the study, as well as an evaluation of its effectiveness across different clinical scenarios [3,5,38].

### 3.3. Computed Tomography Lymphangiography (CTL)

CTL is an advanced technique in veterinary oncology for mapping SLNs, particularly in MCTs. CTL utilizes computed tomography imaging to elucidate the lymphatic system and facilitate SLN mapping. This technique aligns with radiographic lymphography principles, particularly in the use of water-soluble contrast agents. These agents are rapidly absorbed by lymph vessels close to the tumor, reach a peak concentration, and then decrease over time, aiming to promptly delineate the post-contrast administration of SLNs.

The procedure begins with a baseline pre-contrast CT scan using a 64-slice helical scanner, targeting both the tumor and anticipated SLN areas, with images captured at a 2 mm slice thickness using a soft tissue algorithm. For precise SLN mapping, 1 mL of a contrast medium, such as Iohexol, Omnipaque, or Iopamidol, is peritumorally injected. Initial and follow-up scans are conducted at set intervals post injection to pinpoint SLNs or until 20 min pass without SLN detection. Once the contrast appears in an SLN, further scans at 1, 3, and 5 min are performed to ensure the complete visualization of the SLN, concluding the imaging sequence [3,22].

CTL performed on twenty canine MCTs classified as cutaneous (ten out of twenty tumors), subcutaneous (eight out of twenty tumors), and mucocutaneous (two out of twenty tumors) showed a notable 95% detection rate [11]. The CTL mapping technique failed to detect the SLN in only one dog, which had a subcutaneous MCT. The study by Alvarez-Sanchez et al. highlights the potential of CTL; however, the study was performed on only twenty dogs and thus should be interpreted with caution. The study showed that combining CTL and (near-infrared fluorescence) NIR lymphography for MCT SLN mapping resulted in higher detection rates than using either modality alone. Specifically, the detection rate of SLNs using CTL alone was 95% (19 out of 20 dogs), while NIR alone achieved a detection rate of 80% (16 out of 20 dogs). The integration of both CTL and NIR techniques increased the detection rates since each technique detected different SLNs in different detection times; therefore, a combined approach can significantly enhance SLN detection efficiency in comparison to using each modality independently. Despite most MCTs being classified as intermediate to low grade, a high rate of lymph node metastasis was noted, suggesting an unexpected aggressiveness in these tumors. Interestingly, six of eight subcutaneous MCTs had metastatic SLNs, which indicates the aggressive behavior of subcutaneous MCTs. The literature emphasizes the complementary nature of these imaging techniques, enhancing the likelihood of detecting additional SLNs and highlighting the need for comprehensive SLN identification in MCT cases [11].

Grimes et al. (2017) reported a slightly lower 89% success rate in detecting SLNs using CTL in a study involving 18 dogs with various tumor types [22]. The 100% detection rate was for MCTs, but only two cases of undefined MCTs were included. The diversity in tumor types within this study highlights a potential limitation in directly applying the overall success rate specifically to MCTs since only two dogs with MCTs were included in the study. This underlines the need for more focused research to accurately assess the efficacy of CTL in the context of MCTs [22]. In another study by Grimes et al. performed on fifteen dogs, including eleven with pre-diagnosed MCTs, who underwent CT lymphangiography and lymph node removal, fifteen SLNs were extracted from MCT-affected dogs [32]. Following CT lymphangiography, the dogs underwent surgery for the excision of the identified SLNs. Shortly before excision (5–15 min prior), 0.1 mL of methylene blue dye, diluted in saline, was peritumorally injected to facilitate the visual identification of the SLN. The first lymph node that stained blue was excised as the SLN. While CT lymphangiography directed surgeons to the SLN’s lymphocentrum, the addition of the methylene blue dye proved essential for accurately intraoperatively identifying the specific SLN [32], as shown in Figure 2. Notably, in all dogs with pre-diagnosed MCTs, at least one SLN was detected, indicating a high level of efficacy in SLN identification in this group. On the other hand, it demonstrated a lack of correlation between the contrast enhancement of SLNs in CT images and the metastatic status in the histopathological examination of the SLNs.

In summary, the first results for CTL application for canine MCTs exhibit a high effectiveness in MCT lymph node detection; however, further studies including a higher number of animals with MCTs are needed to refine its application in MCTs. Moreover, the use of CTL has several challenges. Animals undergoing CTL are exposed to ionizing radiation and may require multiple scans, leading to higher costs and potential complexities regarding scanning intervals [41].

### 3.4. Near-Infrared Fluorescence (NIR)/Near-Infrared Fluorescent Image-Guided Lymphography (NIR-LND)

NIR utilizes indocyanine green dye to enhance the visualization of lymphatics and lymph nodes. This process involves administering an intraoperative injection of the dye prior to tumor removal. In NIR-LND, the visualization is aided by a hand-held near-infrared camera. In instances where an immediate fluorescent signal is not evident, a gentle massage may facilitate detection. This technique enables surgeons to accurately track the fluorescent-marked vessels, precisely incise over the lymph nodes, and effectively differentiate between lymphatic and adipose tissue during the resection process.

The study by Alvarez and Sanchez reported an 80% detection rate (sixteen out of twenty MCTs) for SLN using NIR alone, as stated above; however, the main limitation of this study is a small number of animals included: only twenty dogs with MCTs [11]. The four cases that failed were subcutaneous MCTs.

NIR-LND demonstrated similar efficacy in the detection of lymph nodes in dogs with MCTs. A retrospective study by Beer et al. (2022) on 35 dogs with MCTs (25 cutaneous, eight subcutaneous, and two undefined) revealed an 83% successful identification rate of NIR-LND, pinpointing 58 out of the 70 planned nodes for resection [34]. This rate was notably higher compared to the traditional method of unguided locoregional lymph node dissection, which identified 50 out of 70 nodes (74%). Additionally, the metastatic node detection rate was 68% (24 of 35 dogs) in contrast to the 33% detection rate (14 of 43 dogs) using the traditional method without imaging guidance. Moreover, NIR-LND consistently detected at least one SLN in every case. Remarkably, no complications from either NIR or NIR-LND were documented [34]. The limitations of NIR and NIR-LND include limited penetration depth and potential challenges in mapping nodes located in body cavities. Nevertheless, its overall effectiveness, particularly in identifying metastatic nodes, underscores its potential in MCT management [35].

In conclusion, NIR lymphography for MCT detection, particularly when applied as NIR-LND, presents significant advantages, including reliability, visualization accuracy, and cost effectiveness, although its limitations should be taken into account when selecting it as a diagnostic tool.

### 3.5. Contrast-Enhanced Ultrasonography (CEUS)

CEUS is recognized as a valuable non-ionizing mapping technique for MCTs, employing ultrasound in combination with specific contrast agents, such as 25% albumin solution and hydroxyethylated starch, as well as formulations such as SonoVue^®^ (Bracco Imaging S.p.A., Milan, Italy), Sonazoid^®^ (GE Healthcare, Chicago, IL, USA), and Definity^®^ (Lantheus Medical Imaging, Inc., North Billerica, MA, USA), [1]. This method demonstrates a high detection rate of 95.2% (in 59 out of 62 dogs with MCTs) [41]. Out of 62 MCTs, 35 MCTs were histopathologically examined and classified as cutaneous (n = 20) and subcutaneous (n = 15). However, it is important to note that, in the study conducted by Fournier et al., only 55% of the identified SLNs underwent histopathological evaluation [41]. Metastasis was detected in 60% of these evaluated SLNs. Interestingly, metastatic disease was identified in 50% RLNs that were not initially identified as SLNs. Fournier et al. acknowledged several limitations in their study, including the challenge in confirming whether the extirpated LNs were those identified by CEUS due to the lack of marking SLNs preoperatively. This could potentially be addressed in future studies by deploying a guidewire or combining CEUS with intraoperative blue dye SLN mapping. Additionally, the selection of lymphatic basins assessed by CEUS was challenging to standardize, which may have influenced the results. The study also highlighted that while CEUS SLN mapping is safe and demonstrates a high SLN detection rate, it was not directly compared with other SLN mapping techniques, such as lymphoscintigraphy or blue dye injection. Furthermore, some SLNs might have been second-order lymph nodes, and the enhancement patterns only had a moderate agreement with the histological metastatic status of SLNs [41].

In conclusion, CEUS presents a promising tool for SLN mapping in dogs with MCTs, with its safety and high detection rate. However, the technique’s accuracy in predicting metastatic status and the challenges in identifying draining lymph nodes correctly underscore the need for further research, potentially combining CEUS with other mapping techniques.

## 4. Conclusions

In summary, lymph node mapping, resection and lymph node staging are becoming increasingly recognized as pivotal elements in the surgical oncology of canine MCTs. Given the propensity of MCTs to metastasize to lymph nodes, an accurate nodal assessment is crucial for effective staging and subsequent treatment planning. SLN mapping techniques are emerging as essential tools to improve the precision of MCT staging. The development and refinement of these techniques aim to standardize lymphadenectomy procedures for MCTs, as well as potentially enhancing therapeutic outcomes. Lymphoscintigraphy has the highest lymph node detection rate in MCTs (up to 97.9–100%). It is currently defined as a gold standard for lymph node mapping both in human and veterinary medicine. Additionally, the first studies on a limited number of animals revealed that the advanced technique of CTL has demonstrated significant efficacy in SLN mapping for MCTs (detection rate up to 95–100%), offering non-ionizing, high-detection alternatives, although it comes with its unique set of challenges and limitations. The similar detection rate was assessed in the first study on the CEUS application performed on 65 canine MCTs, making it a promising technique. Combining both CTL and NIR/NIR-LND also yields promising results for SLN detection in canine MCTs as each technique enables the detection of different SLNs. The integration of these mapping techniques into routine MCT staging is crucial, but further clinical trials on a large number of animals are needed to refine procedures and fully assess the clinical benefits of each technique.

## Figures and Tables

**Figure 1 animals-14-01089-f001:**
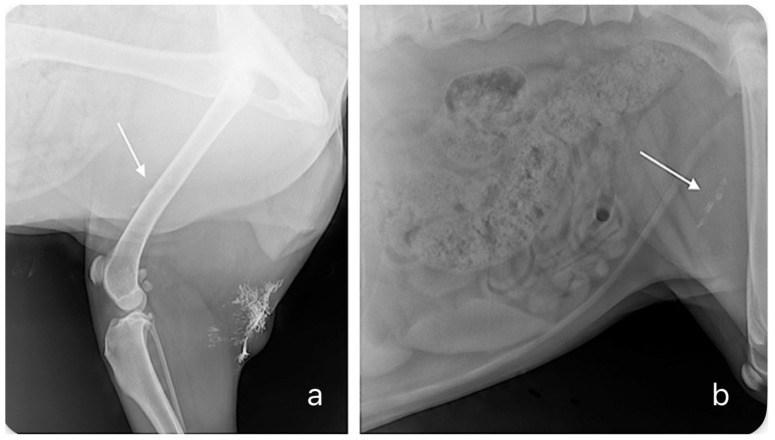
Radiographs presenting indirect lymphography 24 h after the peritumoral injection of Lipiodol in a dog with a tumor located in the popliteal region. (**a**) Lateral radiograph of the caudal part of the abdominal wall and popliteal region. A shadow in the popliteal region corresponds to the site of the injection of the contrast agent. The arrow indicates a superficial inguinal lymph node. (**b**) A radiograph that better visualizes the superficial inguinal sentinel lymph node (arrow).

**Figure 2 animals-14-01089-f002:**
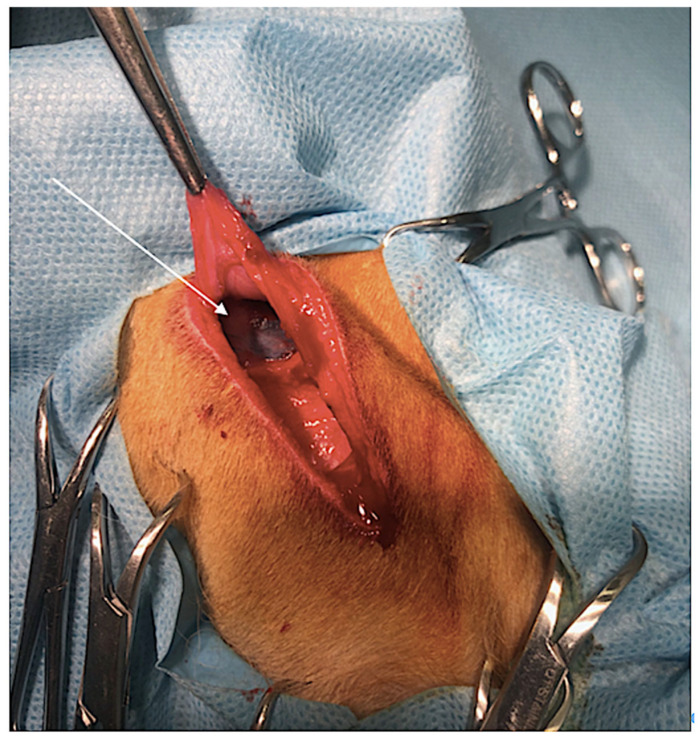
Intraoperative view of the ventral aspect of a neck after a four-quadrant injection of methylene blue into the tumor area; the tumor is located on the upper lip. The blue dye flows along with the lymph from the tumor masses to the SLN and the mandibular lymph node, staining it blue (arrow). This allows for the easier identification, dissection, and removal of the SLN.

**Table 1 animals-14-01089-t001:** Clinical staging system for canine cutaneous MCT proposed by the WHO.

Clinical Staging System for Canine Cutaneous MCT Proposed by the WHO	
Stage	Description
0	A single tumor incompletely excised from the dermis, histologically identified, without regional lymph node involvement a. Without systemic signs b. With systemic signs
I	A single tumor confined to the dermis, without regional lymph node involvement a. Without systemic signs b. With systemic signs
II	A single tumor confined to the dermis, with regional lymph node involvement a. Without systemic signs b. With systemic signs
III	Multiple dermal tumors; extensive, infiltrating tumors with or without regional lymph node involvement a. Without systemic signs b. With systemic signs
IV	Any tumor with distant metastasis, including involvement of blood or bone marrow

**Table 2 animals-14-01089-t002:** Comparative analysis of sentinel lymph node mapping techniques used in canine mast cell tumors (MCTs), including the detection rates, specificity, and limitations of each technique.

Technique	Detection Rates ^1^ in MCTs [%]	Research	Number of Dogs	Number and Type of MCTs	Limitations
Lymphoscintigraphy	95%	Worley 2014 [18]	19 dogs	20 MCT (both sMCTs and scMCTs): − unspecified number of individual MCTs	There is a reliance on radioactive isotopes, elevated costs, and risks of radioactive environments.
	91%	Ferrari et al., 2020 [9]	30 dogs	34 MCTs: − 24 cMCTs, − 6 scMCTs, − 3 scars ^2^ − 1 undefined ^3^	
	100%	Ferrari et al., 2021 [21]	53 dogs	66 MCTs: − 50 cMCTs − 16 scMCTs	
	100%	Chiti et al., 2021 [36]	23 dogs ^4^	8 MCTs: − 4 cMCTs, − 2 scMCTs − 2 mcMCTs	
	97.9%	Manfredi et al., 2021 [37]	59 dogs ^4^	47 MCTs: − undefined ^3^	
	100%	Gariboldi et al., 2022 [4]	33 dogs ^4^	19 MCTs − 14 cMCTs: − 4 scMCTs scars − 1 mcMCT scars	
Colorimetric SLN mapping using methylene blue	90%	Worley 2014 [18]	19 dogs,	20 MCTs: − undefined ^3^	Due to its limited sensitivity and specificity, this method is not advised for exclusive use but rather as a supplementary intraoperative tool alongside CT lymphography, lymphoscintigraphy, or indirect lymphography.
	86%	Brissot et al., 2017 [38]	29 dogs ^4^	15 MCTs: − 11 cMCTs, − 3 scMCTs − 1 MCTs undefined ^3^ − 3 scars ^2^	
	91%	Ferrari et al., 2020 [9]	30 dogs	34 MCTs: − 24 cMCTs, − 6 scMCTs, − 3 scars ^2^ − 1 undefined ^3^	
Indirect lymphography	86.6%	Brissot et al., 2017 [38]	29 dogs ^4^	15 MCTs ^2^: − 11 cMCTs: − 3 scMCTs, − 1 MCT undefined ^3^ − 3 scars ^2^	There are inconsistent results when using small volumes of water-soluble contrast, and experiencing a limited depth of penetration. The absence of intraoperative colorimetric techniques in studies using an oil contrast agent may affect the verification of sentinel lymph nodes. Abundant subcutaneous fat could limit the absorption of the contrast agent by lymphatic vessels associated with small tumors.
	90%	De Bonis et al., 2022 [5]	26 dogs	29 MCTs: − 21 cMCTs − 8 scMCTs	
	77.9%	Haas et al., 2023 [39]	53 dogs	59 MCTs, 34 MCTs (diagnostic IL): − 17 cMCTs − 4 scMCTs − 13 undefined ^3^	
	95%	Annoni et al., 2023 [40]	80 dogs	138 MCTs: − 114 cMCTs − 23 scMCTs − 1 mcMCTs	
Computed tomography lymphangiography (CTL)	100%	Grimes et al., 2017 [22]	18 dogs ^4^	2 MCTs: − undefined ^3^	This entails exposure to ionizing radiation and may require multiple scans. This is a lack of colorimetric intraoperative methods to further confirm SLNs. A small number of animals in each study were used (from only two to approximately twenty).
	100%	Grimes et al., 2020 [32]	15 dogs ^4^	11 MCTs: − 10 cMCTs − 1 scMCTs	
	90%	Lapsley et al., 2021 [19]	17 dogs	20 MCTs: − 13 cMCTs − 7 scMCTs	
	95%	Alvarez-Sanchez et al., 2022 [11]	20 dogs	20 MCTs: − 10 cMCTs − 8 scMCTs − 2 mcMCTs	
Near-infrared fluorescence (NIR)/ near-infrared fluorescent image-guided lymphography (NIR-LND)	83%	Beer et al., 2022 [34]	35 dogs	35 MCTs: − 25 cMCTs − 8 scMCTs − 2 undefined ^3^	There is a limited penetration depth, and there are challenges in mapping nodes in body cavities.
	80%	Alvarez-Sanchez et al., 2022 [11]	20 dogs	20 MCTs: − 10 cMCTs − 8 scMCTs − 2 mcMCTs	
	100%	Gariboldi et al., 2022 [4]	33 dogs ^4^	12 MCTs scars: − 7 cMCTs scars − 5 scMCTs scars	
Contrast-enhanced ultrasound (CEUS)	95.2%	Fournier et al., 2020 [41]	59 dogs	62 MCTs: − 20 cMCTs − 15 scMCTs − 27 undefined ^3^	This method offers a high detection rate and is effective in identifying SLNs in MCTs.

cMCTs—cutaneous MCTs; scMCTs—subcutaneous MCTs; mcMCTs—mucocutaneus MCTs. ^1^ Detection rate—the percentage of MCTs for which at least one SLN was successfully identified using a specific diagnostic technique. ^2^ Scar—after incomplete MCT resection; included by the authors of the study into MCT group. ^3^ Undefined—no histopathology report available. ^4^ Dogs presenting variable tumor types, among which MCTs are included.

## Data Availability

Not applicable.

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
