# Peer review of "The Use of Sentinel Lymph Node Mapping for Canine Mast Cell Tumors"

_animals, 2024, doi:10.3390/ani14071089_

Round 1

Reviewer 1 Report

Comments and Suggestions for Authors

In the manuscript the authors provided a review of the published studies describing the application of sentinel lymph node (SLN) identification in canine mast cell tumors (MCT). The review can be useful for clinicians, surgeons and oncologists and can set the basis for planning new studies on this topic. Currently, at the diagnostic level (clinical and cytological) and in the prognosis definition there are some difficulties regarding the subcutaneous MCT. According to the existing evidence, subcutaneous MCT should be viewed as a different neoplastic presentation and the tumors are not grading with the same criteria used in cutaneous MCT. The distinction between subcutaneous and cutaneous was not usually performed in first survival studies and case series of MCT.

In the manuscript there is a lack of description of subcutaneous MCT. The authors should include in the introduction a subsection regarding clinical, histological and behavior features of subcutaneous MCT. I suggest that it should be included before subsection of the grading.

In the detailed explanation of each SLN methodology, the authors should clearly state if the study differentiate cutaneous from subcutaneous MCT.

Table 2 should be improved by include the number of cases of each study and again if the MCT included were cutaneous or subcutaneous.

Moreover, a brief paragraph about the prognostic and predictive factors of MCT such as Kit and Ki67 (MIB index) should be included before the description of the methods of mapping the SLN.

Other points:

Summary

1) not all MCT are malignant. So, I suggest to remove malignant and put only cutaneous and subcutaneous.

2) correct to “by spreading”

3) last sentence, correct to “¾ could be a notable progression in the veterinary oncology”.

Introduction

4) “The majority of failures occur due to local/regional disease, suggesting potential benefits from additional therapies like radiotherapy or extensive surgery for MCTs >3 cm”.  Very confusing sentence. Please reformulate.

5) According to the authors the Patnaik system is recognized as the "gold standard". I am not sure if we can use this terminology for the grading. The official classification of mast cell tumors (endorsed by WHO) should be for referring the recommend grading system.

6) “However, this approach, particularly for MCT grade 3, has been debated in various studies [3]”. Only a study is cited. Please cited all the original studies.

7) As stated previously, introduction lacks information about subcutaneous MCT. Before the description of the histological grade system, the authors should refer the histologic difference between cutaneous and subcutaneous MCT. The histologic features used to assess the potential aggressiveness of the subcutaneous MCT (mitotic count)

8) If clinical staging is only for cutaneous MCT this should be added to the text and not only in the table 1.

9) the authors should not use “Patients” instead of animals.

10) The paragraph of the Marconato study is confusing and should be reformulate.

11) in this paragraph “lack of routine c-kit/Ki67 assessment, what can potentially influence the study’s findings”. This information is difficult to understand without a previous explanation on what is c-kit or Ki67.

12) Please further explain this sentence: Furthermore, the research conducted by Ferrari et al. highlights the significant link between subcutaneous MCTs and overt metastasis, underscoring the importance of considering both tumor type and size in SLN mapping strategies.

13) “8 purpose healthy dogs.” What does purpose means?”. I suggest to remove.

Comments on the Quality of English Language

Despite I am not a native speaker, I think the manuscript should be revised regarding the language. I made some suggestions for improvement by a wide revision is needed.

Reviewer 2 Report

Comments and Suggestions for Authors

In this article, Romanska et al. are summarizing the various lymph node mapping methods for canine mast cell tumors. The authors provide a very comprehensive analysis of each method, discussing strengths, weaknesses, challenges and limitations for lymph node detection. This article can be a great tool for veterinary clinicians; therefore, I recommend it for publication.

Minor comment:

-What do the authors mean by "purpose healthy dogs" under 4.2, page 9?

Comments on the Quality of English Language

Minor grammatical editing is required.

For instance, the legend of Figure 2 should be edited as follows: "...indirect lymphography 24 hours after peritumoral injection..."

Reviewer 3 Report

Comments and Suggestions for Authors

This manuscript aims to review the current state of the art regarding methods for identifying sentinel lymph nodes in dog mast cell tumors. Several aspects must be improved:

Remove table 1, no need to included information that is widely available in the literature. Also, the MCT staging is highly controversial, so this table doesn’t bring additional clarification neither you do a solid discussion on this matter.

In page for you are writing about draining lymph node and use the acronym SLN which means sentinel lymph node. You should introduce the term sentinel lymph node, explain what it is and then use the acronym. Otherwise is confuse.

Search methodology needs improvement. How many papers were retrieved? when do you initiate the search and when do you finished it? I found (in a very quick search) additional papers not included in your study, specifically about this topic:

De Bonis A, Collivignarelli F, Paolini A, Falerno I, Rinaldi V, Tamburro R, Bianchi A, Terragni R, Gianfelici J, Frescura P, Dolce G, Pagni E, Bucci R, Vignoli M. Sentinel Lymph Node Mapping with Indirect Lymphangiography for Canine Mast Cell Tumour. Vet Sci. 2022 Sep 8;9(9):484.

Gariboldi EM, Ubiali A, Chiti LE, Ferrari R, De Zani D, Zani DD, Grieco V, Giudice C, Recordati C, Stefanello D, Auletta L. Evaluation of Surgical Aid of Methylene Blue in Addition to Intraoperative Gamma Probe for Sentinel Lymph Node Extirpation in 116 Canine Mast Cell Tumors (2017-2022). Animals (Basel). 2023 Jun 2;13(11):1854.

Annoni M, Borgonovo S, Aralla M. Sentinel lymph node mapping in canine mast cell tumours using a preoperative radiographic indirect lymphography: Technique description and results in 138 cases. Vet Comp Oncol. 2023 Sep;21(3):469-481.

In item 3. System HN0 to HN3: include additional information. HN0 for example, is for cases of lymph nodes with less than 3 isolated mast cells by 10 HPF. Is the authors decided to include this information here, then they should be precise, otherwise information is not useful.

Same item, explain the meaning of this sentence or remove it: For both groups the median disease free interval (DFI) was not reached during the 2 year lasting study.

Same item,: “Notably, dogs with recurrence had clean surgical margins.” remove the sentence or explain why do you think this is a notable information and what do you want to say with this.

Figure 1: the figure can be improved. The legend should also include a summary of the methodology. If you describe 4 injections you should create and image as much as possible, descriptive of the methodology.

Table 2 in incomplete. Additional studies are missing. It is relevant to include the contrast used, when adequate. Also, even if you include the name of the authors you should also include the number of the respective reference. Consider include additional data on it (number of animals, secondary effects, etc…)

Comments on the Quality of English Language

Adequate

Reviewer 4 Report

Comments and Suggestions for Authors

This is an interesting topic and I think a review is well-warranted on sentinel lymph node mapping in mast cell tumors. However, the structure and presentation of the information in this manuscript is hard to follow. In general, I would recommend making each section that discusses SLN mapping (4.1 to 4.5) follow a similar structure. For example, a definition and introduction to the procedure, a general outline of steps, equipment and timeline involved, followed by a review of the literature and then some concluding remarks. Currently, this information is presented in different orders in each section. Also, more of a background on SLN in humans in the introduction would set the stage well for this review. Please see below for specific examples. 

1. Introduction

- Opening sentence: "Cancer is the second main cause of death in small animals." There is no citation and it is not clear what is meant by a "main" cause of death or which animals are included in "small animals". This sentence should be removed or clarified with a citation. 

-In the same paragraph, the first of the introduction, sentences 2, 3 and 4 seem out of order. I would re-order it so that the sentence "Evaluation of locoregional lymph nodes, known as lymph node mapping, is a critical process in assessing the stage of various solid tumors such as: mast cell tumor (MCT), anal gland anal sac adenocarcinoma, melanoma and mammary gland adenocarcinoma" comes after the two sentences that explain the definition of staging.

-3rd paragraph: The sentences regarding primary tumor size seem out of place. It is in the middle of information on why lymph node status is important to prognosis which flows well into the rest of the manuscript. I would recommend removing the following sentences; alternatively if they are important to the authors re-structuring the paragraph so all the information on lymph node status is together, and then talking about primary tumor size separately.

---- Furthermore, dogs with tumor sizes less than 2 cm demonstrated a median PFS of 255 days and OS of 250 days, while those with tumors larger than 3 cm presented with substantially lower median PFS of 41 days and OS of 176 days, underscoring the impact of tumor size on survival outcomes [12].

----The majority of failures occur due to local/regional disease, suggesting potential benefits from additional therapies like radiotherapy or extensive surgery for MCTs >3 cm

-3rd paragraph: The intent of this sentence is not very clear. I think it is meant to indicate that the regional lymph node may differ from the sentinel lymph node due to these new lymphatic connections, which could impact prognosis if the wrong lymph node was evaluated. This sentence should be clarified or rewritten if possible.  "Evaluation of lymph nodes, especially [the] first node in the lymphatic chain draining the tumor, is crucial in the staging process of MCTs, especially since new lymphatic connections can form in MCTs, making this assessment pivotal for accurate prognostication [13-16]."

Section 1.1:

- "It encompasses well-differentiated neoplastic mast cells in Grade 1 to mast cells with significant pleomorphism in Grade 3." This seems to be an incomplete summary of the Patnaik grading scheme. It is a good idea to briefly summarize the grading schemes, but it is recommended to  briefly mention the criteria that are looked at in the Patnaik grade (mitotic figures, karyomegaly, etc.) instead of just mentioning the differentiation. 

- This is more of an opinion, but it is not clear to me that Patnaik is considered the gold standard any longer. 

Section 1.3:

-"Importantly, it classifies dogs with stage II lymph node metastasis or multiple tumors as stage III, regardless of lymph node metastasis." This sentence is not clear. Is it meant to say, "...it classifies dogs with lymph node metastasis as stage II"?

Section 3

-Overall I do not think section 3 is needed for this review. It would have been preferable to get at least a brief background in SLN in humans versus a review of the impact of nodal status on prognosis. If the authors wish to keep this section in please see below for comments.

-These sentences do not make sense with each other. The first sentence refers to dogs with excised nodal metastasis, and the second gives MSTs for different nodal status (HN status): 

----"Emphasizing the role of nodal status in prognosis, a study performed by Weishaar and collaborators on 41 dogs with a total of 48 MCT showed that patients with routinely excised nodal metastases experience longer survival and decreased recurrence rates compared to those without nodal intervention [21]. Specifically, for patients classified as HN0/1the median survival time (MST) was 1,824 days, whereas for those classified as HN2/3, the MST was 804 days."

Similar for these sentences. How do survival rates for the HN status indicate that excising nodal metastasis improves survival?

----"For both groups the median disease free interval (DFI) was not reached during the 2 year lasting study. The survival rates for HN0/1 and HN2/3 patients were 90% and 56%, respectively, indicating that patients with routinely excised nodal metastases tend to have longer survival and lower recurrence rates compared to those without nodal intervention [21]"

-The paragraph starting with "On the other hand, the study performed by Marcano..." needs to be fully revised. The sentence structure makes it hard to understand what is being said (for example- dogs with H2N nodes had a worse OS compared to what group? What is the immune implication mentioned?). The author's name is wrong. 

Section 4:

In general the order of mapping techniques should be kept constant. The list in paragraph 1, Table 1, and the subsequent sections should follow the same order.

-1st paragraph: Please define what is meant by diagnostic rate (for example- identification of the SLN?). There are 6 methods listed here, but only 5 in the table and subsequent sections. Please clarify why colorimetric SLN mapping is not included in the table and does not receive a section of its own. Additionally, only colorimetric SLN mapping gets a brief definition in parentheses, this should be removed for consistency. 

Figure 1: This figure does not add anything useful to the manuscript. Is the four quadrant method always used? If so, then adding some indication in the figure where the injections are taking place might improve it. 

Table 2: Define what is meant by detection rate. Define how you are categorizing the specificity. As mentioned it would make sense to have the order of the rows in the same order as the following sections.

Section 4.1- Methylene blue or blue dye is referenced several times in this section. However, no introduction to this method has been provided. Why does this method not have a section of its own?

-Please provide a definition of planar imaging. 

-It is mentioned that this is the gold standard. Does this mean in veterinary medicine? In people? And for what tumor types?

-The discussion regarding Chiti et al., is not clear- when the blue dye is referred to, is this considered a gold standard to compare to? Or this study was comparing different methods?

-What is meant by "...although SLNs were not identified in tumors with previous scar tissue."? It is not clear this means recurrent tumors, although I think that is what is meant.

-The paragraph starting with "Furthermore, the research conducted by Ferrari et al. highlights the significant link between subcutaneous MCTs and overt metastasis..." is out of place in this section on lymphoscintigraphy. I would recommend removing it or moving it to the introduction if the authors want to keep it in.

- It seems that one of the citations is wrong at the end of this section- there is no reference to the human literature after the claim that it is the gold standard in people.

Can you explain what a "failure" means? No uptake, multiple nodes highlights, etc.?

Section 4.2

-This section should start with a definition of the procedure, including an overview of the steps and timeline involved.

-What is considered a non-diagnostic observation in this procedure?

-Please include a description of what Lipiodol is.

The remaining sections contain similar concerns- largely a lack of description of the techniques.

Comments on the Quality of English Language

There are several places throughout the manuscript where an article (such as "a") is left out. For example in the Simple Summary, it reads "Lymph node mapping is routinely applied diagnostic technique...", while it should read "Lymph node mapping is a routinely applied diagnostic technique...".  

There are other minor errors throughout the manuscript that should be fixed as well (for example, in section 1.1 "The Patnaik Grading System is categorizing MCTs into three grades." should read "The Patnaik Grading System categorizes MCTs into three grades.

Another example near the end of section 1 " In veterinary medicine, the first approaches to utilize SLN mapping has been performed for various neoplasia, including MCTs [1,20]." should read " In veterinary medicine, the first approaches to utilize SLN mapping have been performed for various neoplasia, including MCTs [1,20]." 

There are other examples throughout the paper that should be fixed.

Round 2

Reviewer 1 Report

Comments and Suggestions for Authors

This revised version of the manuscript was improved. The description of the SLN mapping techniques with the discrimination between subcutaneous and cutaneous MCT clearly augments the interest of the review for all readers and especially for clinicians and oncologists.

Few minor points need revision.

Summary

Line 12: Node is missing

Line 23: why tree quarters?

Abstract

Line 29 remove canine.

Introduction

Line 78: the acronym MCT should be used always throughout the text.

Line 104 “distinct patterns” what does mean? Could it be growth? As to the mitoses, in the diagnostic setting the parameter that usually is used is the mitotic count (number of mitoses per a defined number of high-power fields)

Line 107: I recommend that instead of “particularly subcutaneous” the authors use including subcutaneous.

Lines 110, 111: I suggest “for therapeutic decisions.”

Line 196: this reference (Halsted W, 1894). should be included in reference list and cited in text as a number.  

Line 220: it should be January 2024 I guess.

Would be the figure 1 removed? The legend is deleted.

Legend of Table 1: the word cutaneous should be corrected;

Line 497: reformulated for: “only mild peritumoral, post-injection swelling noted in some cases”.

Line 547: remove the parenthesis MCTs (classified

Line 564: “Interestingly, six of eight subcutaneous MCTs had metastatic  SLNs, which indicates the aggressive behavior of subcutaneous MCTs.” This is a bold statement. I would be more cautious and I suggest to put something like this: Interestingly, six of eight subcutaneous MCTs had metastatic 563 SLNs, which indicates that at least some subcutaneous MCTs can have an aggressive clinical presentation.

Comments on the Quality of English Language

Clearly improved. I suggest very minor corrections (despite I am not a native speaker).

Reviewer 4 Report

Comments and Suggestions for Authors

Overall this revised version of the manuscript is very well done. The flow is much better and the presentation of the material is clear. Please find below very minor suggestions.

Lines 57-58: The cited paper does not support the statement that cancer is the leading cause of death in domestic dogs. It only says in the introduction that cancer is "a" leading cause of death, meaning it is somewhere near the top 

69-70: In the order of most commonly diagnosed cancers, is MCT before or after mammary carcinoma? As it reads, it sounds like MCT is second and mammary is third.

172- add "distant" prior to metastasis to distinguish from lymph node metastasis.

Comments on the Quality of English Language

Line 15- Add "a" to "Lymph node mapping is a routinely applied..."

Lines 22-24- This portion does not make sense, the sentence seems complete without it: "...the successful diagnosis of three-quarters of these tumors could lead to significant progress in veterinary oncology."
